# An Investigation on the Thermally Induced Compatibilization of SBR and α-Methylstyrene/Styrene Resin

**DOI:** 10.3390/polym13081267

**Published:** 2021-04-13

**Authors:** Arnaud Wolf, João Paulo Cosas Fernandes, Chuanyu Yan, Reiner Dieden, Laurent Poorters, Marc Weydert, Pierre Verge

**Affiliations:** 1Luxembourg Institute of Science and Technology, L-4362 Esch-sur-Alzette, Luxembourg; arnaud.wolf@list.lu (A.W.); joao.cosas@list.lu (J.P.C.F.); chuanyu.yan@list.lu (C.Y.); reiner.dieden@list.lu (R.D.); 2University of Luxembourg, L-4365 Esch-sur-Alzette, Luxembourg; 3Goodyear Innovation Center, L-7750 Colmar-Berg, Luxembourg; laurent_poorters@goodyear.com (L.P.); marc_weydert@goodyear.com (M.W.)

**Keywords:** compatibilization, rheology, atomic force microscopy, solid-state NMR

## Abstract

The miscibility between two polymers such as rubbers and performance resins is crucial to achieve given targeted properties in terms of tire performances. To this aim, α-methylstyrene/styrene resin (poly(αMSt-*co*-St)) are used to modify the viscoelastic behavior of rubbers and to fulfill the requirements of the final applications. The initial aim of this work was to understand the influence of poly(αMSt-*co*-St) resins blended at different concentrations in a commercial styrene-butadiene rubber (SBR). Interestingly, while studying the viscoelastic properties of SBR blends with poly(αMSt-*co*-St), crosslinking of the rubber was observed under conditions where it was not expected to happen. Surprisingly, after the crosslinking reactions, the poly(αMSt-*co*-St) resin was irreversibly miscible with SBR at concentrations far above its immiscibility threshold. A detailed investigation involving characterization technics including solid state nuclear magnetic resonance led to the conclusion that poly(αMSt-*co*-St) is depolymerizing under heating and can graft onto the chains of SBR. It results in an irreversible compatibilization mechanism between the rubber and the resin.

## 1. Introduction

The specifications of applicative materials are generally obtained by blending polymers of different chemical and physical characteristics [1,2,3,4]. Their miscibility is a crucial parameter to reach the targeted properties [5,6,7]. In the tire industry, rubbers are particularly subjected to an adjustment of their viscoelastic properties by their blending with another polymer [8,9,10]. Among them, poly(styrene-*co*-butadiene) rubber (SBR, Scheme 1A) is the main component of tire tread formulations thanks to its relatively high glass transition temperature (T_g_) compared to other elastomers, as well as the temperature dependence of its loss factor [8,11]. Nevertheless, a modification of its viscoelastic properties is needed to fit the wet and dry grip requirements and to maximize its performance. Its blending with performance resins is a way to reach these specifications [12].

Performance resins regroup amorphous materials with a low average molecular weight (between 800 and 4000 g/mol). Three groups of synthetic performance resins can be found [13,14]: (i) Resins derived from naturally occurring terpenes produced from monomers isolated from pine trees, (ii) Coumarone-indene resins produced from monomers obtained by distillation of coal tar, and (iii) Aliphatic C_5_ and aromatic C_9_ hydrocarbon resins produced from monomers obtained via steam cracking of petroleum streams. These resins are widely used due to their efficiency to modify the viscoelastic behavior of rubbers to fulfil the requirements of the final applications [15,16,17,18,19,20,21,22].

Copolymers of α-methylstyrene and styrene (poly(αMSt-*co*-St)) belong to the group of aromatic C_9_ hydrocarbon resins (Scheme 1B) [13]. In spite of the extensive use of these resins, the literature is not rich in studies describing the resin concentration influence on the viscoelastic behavior of their blend with a rubber. The initial aim of this work was to fill this gap by assessing the viscoelastic properties of SBR and poly(αMSt-*co*-St) blends at different resin concentrations, from 25 to 150 parts per hundred rubber (phr). As the thermodynamic properties of polymer blends is highly dependent on the molecular weight [4,23,24,25], tailor-made and well-defined poly(αMSt-*co*-St) resins were prepared to avoid any side-effects from the molecular dispersity of the resin [26].

Commercial SBR formulations are known to be mixed with an antioxidant system to prevent their thermal degradation and self-crosslinking [9]. Interestingly, while studying the viscoelastic properties of SBR blends with poly(αMSt-*co*-St), crosslinking of the rubber was observed under conditions where it was not expected to happen. When a polymer crosslinks while dissolved in a second polymer, it leads to a phase separation. This phenomenon, called reaction-induced phase separation (RIPS), results in a narrowing of the miscible concentration range, together with the increase of the polymerization or crosslinking density [27,28,29,30,31,32,33,34]. Surprisingly, after the crosslinking reactions, the poly(αMSt-*co*-St) resin was miscible with SBR at concentrations far above the immiscibility threshold.

This work reports on the investigation of this behavior and attempts to provide an explanation.

## 2. Materials and Methods

### 2.1. Materials

For the resin synthesis, Styrene ≥ 99% (St, Sigma Aldrich) and α-methylstyrene ≥ 99% (αMSt, Sigma Aldrich BVBA, Diegem, Belgium) were distilled over calcium hydride (CaH_2_) under reduced pressure (~30 mbar) prior to reaction. Tetrahydrofuran (THF), cyclohexane and toluene (SLR grade, ThermoFisher Acros BVBA, Geel, Belgium) were purified through an MBraun SPS solvent purification system (alumina and 4 Å sieves columns). The solvent drums were degassed with flowing nitrogen (Alphagaz 2, Air Liquide, Rodange, Luxembourg) at least 30 min prior to the reaction. THF was distilled over sodium/benzophenone (Na(0), ≥99.8% oiled sticks in aluminum foil, and benzophenone, 99% ReagentPlus ^®^, Sigma Aldrich BVBA, Diegem, Belgium) by refluxing while a purple color was observed. Methanol (MeOH, ≥99.8% ExtraDry over molecular sieves, stored under argon, Acros organics BVBA, Geel, Belgium) and *n*-BuLi (2.5 M in hexane, stored under argon at 4 °C, Acros Organics BVBA, Geel, Belgium) were used as received.

For the resin/rubber blend preparation, styrene-butadiene rubber (SBR, Sprintan SLR4602, Trinseo BVBA, Luxembourg, Luxembourg) was used as received. Tetrahydrofuran (THF, SLR grade, ThermoFisher Acros BVBA, Geel, Belgium) was purified through an MBraun SPS solvent purification system (alumina and 4 Å sieves columns). Dicumyl peroxide (DCP, 98%, Sigma Aldrich BVBA, Diegem, Belgium) was purified by recrystallization from MeOH prior use. Hydroquinone (ReagentPlus^®^ ≥ 99%, Sigma Aldrich BVBA, Diegem, Belgium) was used as received.

For the swelling test, toluene (SLR grade, ThermoFisher Acros BVBA, Geel, Belgium) was passed through an MBraun SPS solvent purification system (alumina and 4 Å sieves columns) prior to use.

### 2.2. Synthesis of Poly(αMSt-co-St) Resin

Poly(αMSt-*co*-St) resin with a molecular weight of 1820 g/mol was prepared via anionic copolymerization using the equimolar aliquot addition method, following a procedure reported previously [26]. Molecular characteristics are described in the Appendix A.

### 2.3. SBR/Poly(αMSt-co-St) Blending Procedure

Poly(αMSt-*co*-St) resin was mixed with SBR rubber via solution mixing. A summary of the blend’s composition is described in the Appendix A. The preparation of SBR/poly(αMSt-*co*-St)_150_ is given as an example. Approximatively, 0.12 g of synthetized poly(αMSt-*co*-St) and 0.08 g of SBR (60:40 mixture, 150 phr) were weighed and dissolved in 5 mL of THF until complete homogenization. For rheological measurements, the solution was casted onto a poly(tetrafluoroethylene) (PTFE)-covered glass Petri dish and dried overnight under the hood at room temperature. This treatment was sufficient to remove the traces of solvent. A rubber film was thus recovered and compressed into a rubber ball.

### 2.4. Rheological Mesurements

The rheometer used was an MCR302 equipped with a heating chamber CTD450L (Anton Paar, Graz, Austria). The lower plate was a disposable PP25 and the upper plate was a PP8 (SN50810) measuring system from Anton Paar. The blend was first loaded at 25 °C under the parallel plates and squeezed to have a gap of 1 ± 0.3 mm. The blend was then trimmed to avoid any side effects during the measurements. For the rheological measurements of SBR/poly(αMSt-*co*-St) blends at different resin concentrations, the sample was first heated up to 160 °C and stabilized during~1 min, to remove any internal constraints due to the solvent casting method. The measurement started directly after the stabilization following a 5 °C/min cooling ramp until −50 °C. The applied shear during measurement was following a linear ramp from 1 to 0.02% from 160 to −10 °C. From −10 °C, the applied shear was set to 0.01% to avoid any ruptures due to the maximum torque limit of the instrument. To characterize SBR crosslinking and resin compatibilization by rheology, the SBR/poly(αMSt-*co*-St)_150_ blend was heated up to 215 °C with a temperature ramp of 5 °C/min. The blend was then cooled down directly after reaching 215 °C following a 5 °C/min cooling ramp until −80 °C. The applied shear during measurement was following a linear ramp from 0.1 to 2% from 25 to 215 °C, and from 2 to 0.1% from 215 °C to −10 °C. From −10 °C, the applied shear was constantly set to 0.02%. Due to thermal expansion of the sample, a dynamic normal force of ±0.25 N was applied to maintain contact with the sample in each case.

### 2.5. Gel Permeation Chromatography Measurements

Gel permeation chromatography was performed using an Agilent 1200 Series (Agilent, Diegem, Belgium) equipped with PLgel mixed-C and Mixed-D columns and with three separate detectors (light scattering, viscosimeter and refractive index RI detector). The samples were prepared according to following procedure. A solution of ±3 mg/mL was prepared by dissolving the polymer in THF (HPLC grade, 99.9%, extra pure, anhydrous, stabilized with 2,6-di-*tert*-butyl-4-methylphenol (BHT). After homogenization, the solution was filtered through an Agilent PTFE 0.2 µm filter to remove any dust or solid impurities. The solution was recovered in a 2 mL glass vial and placed in an automatic sample holder.

Prior to measurement, refractive index (RI) and viscosimeter detectors were purged and zero-tuned. The flow rate was set up at 1 mL/min constantly. The measurement started after the injection of 100 µL of prepared solution. The measurement was stopped after the delayed viscosimeter peak (35 min) to avoid interferences with the next sample. The molecular weight distributions were calculated thanks to a poly(styrene) standard calibration (Agilent EasyVial polymer standard, Diegem, Belgium) going from 100 g/mol to 300,000 g/mol.

### 2.6. Atomic Force Microscopy Measurements

A solution of SBR and poly(αMSt-*co*-St) in toluene was cast on a magnetic stainless-steel disc. The resin concentration was defined at 150 phr of poly(αMSt-*co*-St). The blend was dried overnight under the hood at room temperature. A thin film on the metal disc surface was recovered.

Images of the topography and nanomechanical properties of the samples were acquired using the bimodal AM-FM mode of the MFP-3D Infinity AFM (Asylum Research, Santa-Barbara, CA, USA). Two different setups were used to characterize the morphological changes at different temperatures and the nanomechanical properties at room temperature.

For morphological changes at different temperatures, AC160TS-R3 Olympus cantilevers (Asylum Research, Santa-Barbara, CA, USA) were used, mounted in a high temperature cantilever holder (PEEK). Since the holder is not adapted for quantitative nanomechanical analyses, only qualitative stiffness contrast was obtained. In AM-FM mode, the shift in contact resonance frequency of the second mode allows for the acquisition of complementary stiffness contrast simultaneously with the topography. For measurements at high temperature, a modular heating stage PolyHeater (Oxford Instruments, Gometz-la-Ville, France) was used, designed for high temperature polymer studies from ambient to 400 °C in air. The temperature was varied between 25 °C, 160 °C and 215 °C. Samples was imaged at 25 °C and 160 °C but due to the high mobility of the material at 215 °C, images could not be acquired at this temperature. Area of 10 × 10 μm^2^ were imaged with a resolution of 256 × 256 pixels at a scan rate of 3 Hz.

For quantitative measurements of the nanomechanical properties, AC160TS-R3 cantilevers were mounted on a dedicated AM-FM Probe Holder. Cantilever spring constants were measured as about 23 N/m using the GetReal™ Automated Probe Calibration feature and the first and second resonant frequencies were determined to be 240 kHz and 1.33 MHz, respectively. A relative calibration method was used to estimate the tip radius using a dedicated reference samples kit (Model: PFQNM-SMPKIT-12m, Bruker, Karlsruhe, Germany). The tip radius was adjusted to obtain the proper value of 2.7 GPa for the polystyrene reference, matching the deformation applied on the sample of interest. To ensure repulsive intermittent contact mode, the amplitude setpoint was chosen as A_setpoint_/A_0_~0.5 to 0.7 so that the phase is well fixed below 70°. The reported average and standard deviation values of modulus consider at least 4 and up to 6 images in each sample for representative results.

### 2.7. Nuclear Magnetic Resonance Measurements

The liquid- and solid-state nuclear magnetic resonance (NMR) spectra were measured on a Bruker Avance III HD 600 MHz (proton Larmor frequency) spectrometer (Bruker, Karlsruhe, Germany). All the chemical shifts were referenced to tetramethylsilane (TMS) by referencing the residual d-chloroform signal to 7.26 ppm (liquid state) or setting the adamantane methylene signal to 37.77 ppm (solid state). For liquid-state NMR, ~20 mg of neat SBR was dissolved in ~600 µL of deuterated toluene-d8 (99.6%D, Sigma Aldrich BVBA, Diegem, Belgium) in a 5 mm NMR tube. ^13^C spectra with inverse gated ^1^H decoupling were performed on solution samples with a repetition delay of 6 s and a total accumulation of 10,240 scans. Prior to the swollen solid-state NMR experiments, samples were swollen and washed at least 3 times in 100 ml of toluene. To obtain the solid-state ^13^C NMR spectra, ~7 mg of crosslinked SBR (either with the resin CrR-SBR or with dicumyl peroxide CrDCP-SBR) was swollen by ~24 mg of toluene-d8 in a disposable HRMAS insert (B4493, Bruker), which was inserted into a 4 mm ZrO_2_ rotor and spun at 7 kHz spinning rate on a double-resonance MAS probe. One-dimensional ^13^C direct polarization with high-power decoupling NMR experiments were performed on the CrR-SBR and CrDCP-SBR samples with a repetition delay of 4 s, and a total accumulation of 3288 and 2499 scans, respectively.

## 3. Results

### 3.1. Viscoelastic Characterization of SBR/Poly(αMSt-co-St) Blends at Different Resin Concentrations

The viscoelastic properties of SBR/poly(αMSt-*co*-St) blends at different resin concentrations ranging from 25 to 150 phr were determined by rheological measurements. The blends were first heated up to 160 °C, then cooled down to −50 °C. The storage modulus (G’), loss modulus (G”) and loss factor were determined during the cooling ramp. Figure 1 represents the viscoelastic properties of SBR/poly(αMSt-*co*-St) blends at concentrations from 25 to 60 phr (Figure 1A) and from 60 to 150 phr (Figure 1B) compared to neat SBR (black curves).

The T_g_ of SBR was shifting depending on the resin concentration. Up to 60 phr, poly(αMSt-co-St) is fully miscible in SBR as only one T_g_ is detected. Between 60 and 80 phr, a second glass transition is starting to appear in the range of the resin T_g_ (around 80–120 °C), indicative of a phase separation. From 80 phr, the blend T_g_ is relatively constant and a clear biphasic behavior was observed. This value was taken as the immiscibility threshold of the resin in SBR. The results suggest that SBR is saturated with poly(αMSt-*co*-St) resin, and the excess of resin is phase-separating.

### 3.2. Evidence of SBR/Poly(αMSt-co-St)_150_ Blends Crosslinking and Enhanced Compatibility

SBR loaded with 150 phr of poly(αMSt-*co*-St) was selected to illustrate the SBR crosslinking and enhanced compatibility between both components, as an unambiguous phase separation was observed. When rheological measurements of SBR/poly(αMSt-*co*-St)_150_ were performed while heating up from 20 to 215 °C, an increase of G’ was observed. This trend was not expected since SBR alone did not show such behavior (Figure 2A, red curves). Surprisingly, the G’ increase was the result of the crosslinking of SBR, as attested by swelling measurements of the tested samples immersed in toluene. Indeed, after the test, the sample swelled more than 2000% in toluene (Table 1). From the Flory-Rehner equation, the SBR crosslink density was calculated to be equal to 2.18 × 10^−4^ mol/cm^3^, corresponding to one crosslink node every 85,000 g/mol of SBR. It is noteworthy that SBR alone heated up to 215 °C remains fully soluble in toluene, as well as SBR/resin blends heated up to 160 °C. The crosslinking of SBR in the presence of the resin indicates that a thermally induced reaction occurred during the measurement.

In addition, the measurement of the viscoelastic properties of SBR/poly(αMSt-*co*-St)_150_ heated up to 215 °C exhibited a significantly different behavior than SBR/poly(αMSt-*co*-St)_150_ heated only up to 160 °C. While the later revealed two glass transition temperatures indicating that the immiscibility threshold is passed (Figure 2B, blue curves), the blend heated up to 215 °C only had one relaxation (Figure 2B, black curves). This indicates that, after the thermal treatment at 215 °C, the miscibility of both components has been improved. The stability of this behavior was checked by successively re-measuring the sample three times with successful heating–cooling cycles between −50 and 160 °C (see Appendix A). In each case, just one single relaxation was observed, meaning this reaction was irreversible.

### 3.3. Atomic Force Microscopy Measurements of SBR/Poly(αMSt-co-St)_150_

The phase behavior of SBR/poly(αMSt-*co*-St)_150_ heated at either 160 °C or 215 °C was checked by following the evolution of the morphology by AFM. The control of the temperature was performed using a heating stage under the AFM head, for a duration of 20 min at each temperature.

Figure 3 shows images of SBR/poly(αMSt-*co*-St)_150_ taken at 25 °C and sequentially heated at 160 °C at the same location. At 25 °C (Figure 3A), the stiffness contrast images of the area show homogeneous properties with no phase separation. It was assumed that the absence of phase-separation before the heat treatment was attributed to the solvent-casted blend preparation, which may induce a non-equilibrium state of mixing. At 160 °C, a clear phase separation is revealed in the stiffness contrast. Indeed, the softer SBR matrix appears in dark brown and the stiffer poly(αMSt-*co*-St) resin in light yellow. With a scan line of 3 Hz and 256 × 256 pixels, each image was taken in less than three minutes. In image B, one can see that the size of the phase-separated resin changes from a few hundred of nanometers in the beginning of the image (top) to a few microns at the end of the image (bottom), i.e., the phase separation and coalescence of the poly(αMSt-*co*-St) resin happened during imaging. The same effect can be seen in images C and D, until the system reaches stability, with the phase-separated resin reaching several microns in diameter. The bright, rough particle (probably dust on the surface) is an indication of the thermal drift of the sample under the tip. The last image F was taken with the dust particle as a reference of the location for imaging after treatment, which was retrieved after cooling, confirming that the phase-separation is retained (see Appendix A). The same experiment was repeated to confirm the trends observed. Complementary images (see Appendix A) consistently show the phase separation and coalescence of the poly(αMSt-*co*-St) resin, with sizes varying from ~200 nm to 3 μm, which can be related to the local amount of resin at the imaged location, the thickness of the film and the duration of thermal treatment. These results agree with the rheological measurements performed at 160 °C which were showing the phase separation of poly(αMSt-*co*-St) at the same resin concentration (150 phr).

After proving the phase separation during heat treatment at 160 °C, a fresh sample (also at 150 phr resin loading) was thermally treated from 25 to 215 °C and images were taken at each step to follow the evolution of the morphology, as shown in Figure 4. The insets show representative images of the topography and stiffness contrast taken at each step. At 25 °C (Figure 4A), again no phase separation was seen in the stiffness contrast image. The sample was then heated at 200 °C/min up to 160 °C. The morphology observed on the images taken at 160 °C (Figure 4B) is similar to images recorded before on Figure 3. After slowly cooling back to 25 °C (Figure 4C), one can see by the stiffness contrast that the phase separation is maintained with the poly(αMSt-*co*-St) resin phase appearing as round domains of about 3 μm in diameter. Phase separation is also seen in topography, which is rougher compared to the images recorded before the treatment, following the stiffness difference between the phases. Due to the low viscosity of the material at high temperature, it was impossible to acquire images at 215 °C. Nevertheless, images of the material at 25 °C after the treatment at 215 °C for 20 min (Figure 4D), show no phase separation, with a smooth surface and a homogeneous stiffness contrast, in excellent agreement with the rheological measurements done after the treatment of the samples at 215 °C.

Finally, quantitative nanomechanical measurements were done in the neat materials and on the blend at different steps of thermal treatment. Results are shown in Table 2. It is worth noting that elastomers being viscoelastic materials, their viscoelastic properties are time-dependent and therefore are greatly affected by the frequency of analysis. According to the Time–Temperature Superposition (TTS) principle [35,36], high frequency viscoelastic behavior is equivalent to lower temperatures, as the polymer chains do not have enough time to respond to the solicitation. The AM-FM modulus measurements are taken at ~1.3 MHz (second resonance frequency of the cantilever), therefore the values are higher than the usually reported for elastomeric materials at room temperature [37]. Nevertheless, the enhancement effect of the addition of the poly(αMSt-*co*-St) resin in the viscoelastic properties of SBR is verified. Furthermore, the properties are also affected by the thermal treatments, which might be related to the increasing miscibility of the resin in the SBR matrix and crosslinking. Representative images and property histograms are shown in the Appendix A.

## 4. Discussion

The evolution of the viscoelastic properties of SBR with increasing resin concentration showed that poly(αMSt-*co*-St) possesses an immiscibility threshold around 80 phr. Above this concentration, SBR is saturated by the resin and the latter starts to phase-separate. Interestingly, after a heat treatment at 215 °C, SBR/poly(αMSt-*co*-St)_150_ revealed miscibility far above its immiscibility threshold. Rheological measurements, swelling tests and AFM supported the conclusion that SBR was crosslinking in the presence of poly(αMSt-*co*-St) when heated up to 215 °C and lead to the resin compatibilization. This compatibilization was found to be stable over time.

These behaviors were not expected to happen. Indeed, it is known that antioxidants are added to commercial SBR formulations to prevent their thermal degradation and self-crosslinking [9]. It has been reported that when a polymer crosslinks while dissolved in a blend with a second polymer, it leads to a phase separation [27,28,29,30,31,32,33,34]. This phenomenon, called reaction-induced phase separation (RIPS), results in a narrowing of the miscible concentration range, together with the increase of the crosslinking density, due to the reduction of the entropy of mixing. Surprisingly, the opposite behavior was observed in the case of SBR/poly(αMSt-*co*-St)_150_, suggesting another mechanism was occurring. The following discussion attempts to explain this behavior.

SBR alone is stable up to 225 °C (see Appendix A). However, poly(αMSt-*co*-St) resins are known to depolymerize under thermal conditions [38,39,40], leading to the formation of free-radicals by a mechanism of end-chain scission (Scheme 2A).

The alteration of the structural features of poly(αMSt-*co*-St) before and after a thermal heating of 24 h at 215 °C was shown by the determination of its molecular weight distribution (Figure 5). The gel permeation chromatography (GPC) traces depict a clear reduction of the resin molecular weight together with a significant increase of the dispersity (Ð) due to the thermal treatment, in accordance with a radical scission and depolymerization of the resin. The generation of free radicals from the thermal depolymerization of poly(αMSt-*co*-St) (Scheme 2A) could be the origin of the crosslinking of the rubber during the rheological tests. Indeed, the radical crosslinking mechanism of SBR is a well-known phenomenon [38,41,42,43,44,45]. The free radicals generated by poly(αMSt-*co*-St) thermal depolymerization may initiate the oxidative crosslinking of the rubber (Scheme 2B). This step, also called “oxidative hardening” of the rubber, results in a cascade of crosslinking reactions involving oxygen. The radical on the α-position of the vinyl double bond is also highly reactive toward the unsaturation present in SBR and is able to successively react with the [1,2]- and the [1,4]-butadiene units, to form a crosslinked network, as a results of [1,2]- and [1,4]-addition crosslinking reactions (Scheme 2C).

To prove that the crosslinking reaction follows a radical mechanism in SBR/poly(αMSt-*co*-St), a fresh blend was prepared with 30 phr of hydroquinone as antioxidant (SBR/poly(αMSt-*co*-St)_150_/AO_30_). Figure 6A,B represent the viscoelastic properties of SBR/poly(αMSt-*co*-St)_150_/AO_30_ during the heating and cooling ramps, respectively. The presence of hydroquinone clearly inhibits the crosslinking reaction as G’ was not increasing during the heating ramp (Figure 6A, red curve), as opposed to the blend prepared without (Figure 6A, black curve). This behavior indicates that the crosslinking of SBR was effectively inhibited in SBR/poly(αMSt-*co*-St)_150_/AO_30_. The conclusion was further confirmed by immersing SBR/poly(αMSt-*co*-St)_150_/AO_30_ in toluene, resulting in a complete dissolution, whereas SBR/poly(αMSt-*co*-St)_150_ was swelling. As hydroquinone prevents SBR/poly(αMSt-*co*-St) from crosslinking, it confirms that the mechanism is initiated and driven by free radicals, presumably coming from the thermal depolymerization of poly(αMSt-*co*-St) resin.

Interestingly, the SBR/poly(αMSt-*co*-St)_150_/AO_30_ blend still exhibits a clear phase separation of poly(αMSt-*co*-St) resin, even after the heat treatment at 215 °C. In addition, the resin was extracted from SBR/poly(αMSt-*co*-St)_150_/AO_30_ to assess the influence of the antioxidant on the resin depolymerization and revealed a quasi-identical molecular weight distribution. These results indicate that the compatibilization of SBR and poly(αMSt-*co*-St) is also driven by a radical mechanism and is not only due to a decrease of the resin molecular weight.

This compatibilization could be explained by similar observations performed during the preparation of interpenetrated polymer networks (IPN). IPN are a combination of two polymer networks, where at least one of the two polymers is crosslinked in the immediate presence of the other [46,47]. Several pairs of IPN have been made miscible, for instance by introducing a compatibilizer [48,49,50]. In such a situation, the addition of a compatibilizer generally aims at promoting the interactions at the interface between the two immiscible phases and to move their surface energies toward each other [51,52]. Among the different ways to compatibilize polymer blends or networks, the co-reaction of the two networks, or the partial grafting of one of the polymers to the second one, is a good way to enhance their chemical affinity, and their compatibility as well [4]. This last assumption stands as the most probable explanation of the compatibilization between SBR and poly(αMSt-*co*-St).

In an attempt to determine if the SBR structural features were modified during its crosslinking in the presence of poly(αMSt-*co*-St), solid-state NMR (ssNMR) was performed after the complete removal of the resin by solvent extraction. The crosslinked-SBR induced by the resin was named CrR-SBR. The same experiment was done on an SBR sample crosslinked by using dicumyl peroxide (DCP). The resulting product was named CrDCP-SBR (see Appendix A). Both crosslinked rubbers were compared to the neat SBR (liquid state NMR). Figure 7 shows a zoom of the ^13^C NMR in the aliphatic -CH_2_- carbon region of SBR. As depicted on Figure 7, the fingerprint of the aliphatic carbons of SBR was not affected significantly when the rubber was crosslinked with DCP (blue curve). On the other hand, when the rubber was crosslinked in the presence of the poly(αMSt-*co*-St) resin (red curve), a lot of changes were observed in this region. The major changes are highlighted in the black rectangles. These results point out that the poly(αMSt-*co*-St) resin could be involved in other reactions than initiating the rubber crosslinking reaction. Nevertheless, it is rather difficult to confirm that these changes are related to the grafting of poly(αMSt-*co*-St) fragments onto the SBR chains, but still provide evidence that SBR encountered structural changes which could explain the promoted miscibility between SBR and the poly(αMSt-*co*-St) resin.

In the light of all the elements gathered during this study, it can be concluded that the chemical modification encountered by the SBR chains could be the result of a modification due to the species coming from the depolymerization of poly(αMSt-*co*-St). By their reaction with the chains of SBR, they would modify the affinity of the rubber network toward poly(αMSt-*co*-St), improving the miscibility between the resin and SBR. This reaction could be a typical case of reactive compatibilization observed during the synthesis of interpenetrating polymer networks [4]. To further confirm the enhanced affinity of CrR-SBR compared to neat SBR toward poly(αMSt-*co*-St), CrR-SBR was immersed in a solution containing the equivalent of 150 phr of resin solubilized in toluene. Here, the aim was to reintroduce the resin into the network and to analyze its viscoelastic behavior. It was assumed that if the affinity of the rubber network toward poly(αMSt-*co*-St) was enhanced, it would result in a shift of the immiscibility threshold toward higher resin concentration.

The conditions of the experiments were fine-tuned for the crosslinked network CrR-SBR to be able to trap 150 phr of resin. The material after the re-introduction of the resin was renamed CrR-SBR/poly(αMSt-*co*-St)_150_, for the sake of clarity. Figure 8B represents the viscoelastic properties of the material compared to a normal SBR/poly(αMSt-*co*-St)_150_ blend before (blue curves) and after the thermal treatment leading to the enhanced compatibilization of the resin (black curves).

The T_g_ of CrR-SBR was similar to the neat SBR, indicating a low level of crosslinking density (see Appendix A). In CrR-SBR/poly(αMSt-*co*-St)_150_ (Figure 8A, red curves), the incorporation of the resin leads to drastic changes compared to SBR/poly(αMSt-*co*-St)_150_ after treatment at 160 °C (Figure 8A, blue curves). The viscoelastic behavior of CrR-SBR/poly(αMSt-*co*-St)_150_ is closer to the compatibilized case (Figure 8A, black curves), confirming the enhanced compatibility between the two materials [51,52].

## 5. Conclusions

In conclusion, these results demonstrate that the heating of blends of SBR and poly(αMSt-*co*-St) resins above 215 °C leads to their compatibilization. The reaction follows a radical pathway, resulting from the thermal depolymerization of poly(αMSt-*co*-St) as demonstrated by GPC measurements. The generated free radicals would trigger the grafting of the resin onto the SBR chains, resulting in an improvement of their compatibility, as attested by the shift of the immiscibility threshold from 80 phr to over 150 phr. To the best of our knowledge, such a thermally induced compatibilization has never been reported despite several decades of fundamental studies and industrial utilization of SBR and poly(αMSt-*co*-St) resins.

## Data Availability

Data is contained within the article or
Appendix A.

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
