# Peer review of "An Investigation on the Thermally Induced Compatibilization of SBR and α-Methylstyrene/Styrene Resin"

_polymers, 2021, doi:10.3390/polym13081267_

Round 1

Reviewer 1 Report

Thank you for the opportunity in reviewing the manuscript (polymers-1176106). Interesting results are reported and the manuscript was reviewed for publication in Polymers-MDPI Journal. However, the paper require substantial revision before acceptance and consideration for publication. Few points are -

  1. What do authors mean “self-compatibilization” in title?
  2. Abstract is well written. However, discussion on synergistic effect between (poly(αMSt-co-St) and SBR, discussion on its effect on mechanical properties is missing? Moreover, the solid-state NMR is NOT discussed although used as keyword. Please cross-check.
  3. #Section 2.2, the synthesis of Poly(αMSt-co-St) resin following the preparation as reported in reference 26. As the synthesis is already published. So, what is the novelty in present synthesis method?
  4. The scheme of preparation of the compounds is missing in preparation section and will be highly appreciated if provided.
  5. In Figure 1b, the heat build-up at high temperature increases which is NOT good for tyre applications. Please comment on this point. Moreover, the glass transition temperature is significantly altered. Please comment on this for application prospective point of view.
  6. Please comment on how vulcanization is effected since the authors are testing the specimen as high as above 160 oC which is actually or near the vulcanizing temperature?
  7. One major drawback of the work to tune with tire application is author never used the reinforcing fillers such as carbon black which is actually present in tire. Why CB is NOT used to make the work more promising for tire applications?
  8. In #section 3.3, why authors did not study surface roughness of the tire compounds through AFM which is although important?
  9. In Figure 5, why molecular weight decreases and chain-scission of resin decreases with increasing time of thermal treatment?
  10. Again, In Figure 6b with the addition of AO, the heat build-up at high temperature increases which is NOT good for applications. Please comment on this point.
  11. Conclusion can be improved based on further discussion on factors studied in results and experiments?

Author Response

Reviewer #1

Thank you for the opportunity in reviewing the manuscript (polymers-1176106). Interesting results are reported and the manuscript was reviewed for publication in Polymers-MDPI Journal.

We would like to thank the Reviewer for the nice appreciation of our work.

However, the paper requires substantial revision before acceptance and consideration for publication. Few points are:

  1. What do authors mean “self-compatibilization” in title?

In this work, we report the unexpected compatibilization of SBR and poly(αMSt-co-St) at very high resin loading, far above the resin immiscibility threshold. We demonstrate that this compatibilization is occurring by simple heating. However, we agree the term “Self-compatibilization” is confusing. We proposed to rephrase the title as following: “An investigation on the thermally induced compatibilization of SBR and α-Methylstyrene/Styrene.”

  1. Abstract is well written. However, discussion on synergistic effect between (poly(αMSt-co-St) and SBR, discussion on its effect on mechanical properties is missing? Moreover, the solid-state NMR is NOT discussed although used as keyword. Please cross-check.

Thank you for the nice comment, and for noticing the abstract is too much summarizing the work. We modified its end as following: “Surprisingly, after the crosslinking reactions, poly(αMSt-co-St) was irreversibly miscible with SBR at concentrations far above its immiscibility threshold. A detailed investigation involving characterization technics including solid state NMR led to the conclusion that poly(αMSt-co-St) is depolymerizing under heating and can graft onto the chains of SBR. It results in an irreversible compatibilization mechanism between the rubber and the resin “.

  1. #Section 2.2, the synthesis of poly(αMSt-co-St) resin following the preparation as reported in reference 26. As the synthesis is already published. So, what is the novelty in present synthesis method?

As mentioned by the reviewer, we previously reported the synthesis of alternating of poly(αMSt-co-St) in reference 26. Ref 26 was only dedicated to the synthesis and characterization of these new molecules. Our current work submitted to Polymers is a follow-up and deals with the incorporation of these resins in SBR and the study of the viscoelastic properties of the resulting blend. We do not claim at all the present synthesis method is new.

  1. The scheme of preparation of the compounds is missing in preparation section and will be highly appreciated if provided.

The preparation of the compounds is done by mixing the resin and the rubber in THF. We do not think a scheme is essential to describe this sentence. We respectfully decline the proposition of the Reviewer.

  1. In Figure 1b, the heat build-up at high temperature increases which is NOT good for tire applications. Please comment on this point. Moreover, the glass transition temperature is significantly altered. Please comment on this for application prospective point of view.

We agree with the reviewer. Such a high loss factor at high temperature is not good for tire applications. However, this work does not aim at building tires. It is a fundamental approach. This study is even performed on “green” compounds (uncured) which do not contain any curing packages or fillers (carbon black or silica). It is an investigation of the behavior between two polymeric materials under heating.

The increase of the glass transition temperature on Figure 1B is due to the addition of poly(αMSt-co-St). It is well commented and described in the manuscript:

“ The Tg of SBR was shifting depending on the resin concentration. Up to 60 phr, poly(αMSt-co-St) is fully miscible in SSBR as only one Tg is detected. Between 60 and 80 phr, a second glass transition is starting to appear in the range of the resin Tg (around 80 - 120°C), indicative of a phase separation. From 80 phr, the blend Tg is relatively constant and a clear biphasic behavior was observed. This value was taken as the im-miscibility threshold of the resin in SSBR. The results suggest that SBR is saturated with poly(αMSt-co-St) resin, and the excess of resin is phase-separating.”

We would also like to remind that the role of such performance resins is described in the introduction:

“… a modification of its [SBR] viscoelastic properties is needed to fit the wet and dry grip requirements and to maximize its performance. Its blending with performance resins is a way to reach these specifications [12]. … Performance resins regroup amorphous materials with a low average molecular weight (between 800 to 4000 g/mol). Three groups of synthetic performance resins can be found [13,14]: (i) resins derived from naturally occurring terpenes produced from monomers isolated from pine trees, (ii) coumarone-indene resins produced from monomers obtained by distillation of coal tar and (iii) aliphatic C5 and aromatic C9 hydrocarbon resins produced from monomers obtained via steam cracking of petroleum streams. … Copolymers of α-methylstyrene and styrene (poly(αMSt-co-St)) belong to the group of aromatic C9 hydrocarbon resins. “

  1. Please comment on how vulcanization is effected since the authors are testing the specimen as high as above 160°C which is actually or near the vulcanizing temperature?

This work is performed on binary rubber/resin systems only, composed only of SBR and poly(αMSt-co-St). No curing package is added to the recipe, and in consequence the compounds are not vulcanized.

  1. One major drawback of the work to tune with tire application is author never used the reinforcing fillers such as carbon black which is actually present in tire. Why CB is NOT used to make the work more promising for tire applications?

We agree with the Reviewer. The final application of this work may be related to tires. If the effect of poly(αMSt-co-St) on tire compounds would have been the target of the work, it would have needed a relevant tire recipe, composed of a curing package, processing oils and at least one filler such as carbon black or silica. However, this work is fundamental, and aims at generating knowledge concerning the effect of poly(αMSt-co-St) on the viscoelastic properties of SBR alone.

It is worth mentioning that this investigation, as well as ref 26, were done in the frame of a partnership with the Goodyear Tire and Rubber Company. Two of the authors of this paper are working at the Goodyear tire company. Dr. Marc Weydert is Senior technical advisor (https://www.linkedin.com/in/marc-weydert-222902a/?originalSubdomain=lu) and Dr. Laurent Poorters is Engineer (https://www.linkedin.com/in/laurent-poorters-827a061/). It means that, even if the study is not done on a model tire recipe, it remains of high interest for the tire industries.

  1. In #section 3.3, why authors did not study surface roughness of the tire compounds through AFM which is although important?

The focus of this section was to confirm the phase separation of the rubber/resin blend with increasing temperature and further compatibilization after thermal treatment, which was done by the mapping of the stiffness contrast with temperature and by quantitative nanomechanical measurements. The topography alone is not able to provide all this information and roughness measurements would not be useful as the studied systems are uncured, as discussed before.

  1. In Figure 5, why molecular weight decreases and chain-scission of resin decreases with increasing time of thermal treatment?

Figure 5 depicts the evolution of the molecular weight of poly(αMSt-co-St) when heated. It is illustrating the thermal depolymerization of the resin. We agree with the Reviewer that the arrow labeled with “chain-scission” is confusing and has been removed in the revised version of the manuscript.

  1. Again, In Figure 6b with the addition of AO, the heat build-up at high temperature increases which is NOT good for applications. Please comment on this point.

Figure 6 illustrates that the compatibilization of SBR and poly(αMSt-co-St) is not occurring in the presence of an antioxidant. It confirms the work assumption that the compatibilization is triggered by a radical mechanism. Again, this work does not aim at building tire.

Conclusion can be improved based on further discussion on factors studied in results and experiments?

As suggested by the Reviewer, we have phrased the conclusion in a separate part. The conclusion has been rephrased to avoid confusion, line 429 to 438.

“In conclusion, these results demonstrate that the heating of blends of SBR and poly(αMSt-co-St) resins above 215°C leads to their compatibilization. The reaction follows a radical pathway, resulting from the thermal depolymerization of poly(αMSt-co-St) as demonstrated by GPC measurements. The generated free radicals would trigger the grafting of the resin onto the SBR chains, resulting in an improvement of their compatibility, as attested by the shift of the immiscibility threshold from 80 phr to over 150 phr. To the best of our knowledge, such a thermally induced compatibilization has never been reported despite several decades of fundamental studies and industrial utilization of SBR and poly(αMSt-co-St) resins.”

Reviewer 2 Report

Manuscript presents a proper structure and discussion. The following suggestion must be considered before publication:

  1. Line 156

The blends were first heated up to 160°C to remove any internal constraints due to the solvent casting method, then cooled down to -50°C.

Experimental procedure must be described in “materials and methods”. Please remove the following sentence above “results”.

  1. Line 163

Between 60 and 80  phr, a second glass transition is starting to appear in the range of the resin Tg (around 80 - 120°C), indicative of a phase separation. From 80 phr, the blend Tg is relatively constant and a clear biphasic behavior was observed. This value was taken as the immiscibility threshold of the resin in SSBR. The results suggest that SBR is saturated with poly(αMSt-co-St) resin, and the excess of resin is phase-separating.

The authors claim polymer miscibility based on Tg shifting. But, they have to elucidate:

  1. why the second Tg peak is quite small, for samples >80 phr, once a large amount of performance resin was added in the formulation;
  2. the peak broadening of the first Tg as effect of resin incorporation.

  1. Line 176

When rheological measurements of SBR/poly(αMSt-co-St) were performed while heating up from 20 to 215°C, an increase of G’ was observed. This trend was not expected since SBR alone did not show such behavior (Figure 2A, red curves).

Why crosslinking is unexpected, since in line 52, they mentioned that “commercial SBR formulations are known to be mixed with an antioxidant to prevent their thermal degradation and self-crosslinking [9]”?

  1. Line 197

Viscoelastic properties of samples 80 and 100 phr should be presented, in Figure 2A. It would support the author´s decision to focus the experimental procedure  on sample 150 phr, which was not properly described in the text.

  1. Line 212

Figure 3 shows images of SBR/poly(αMSt-co-St) taken at 25°C and sequentially  heated at 160°C at the same location. At 25°C (Figure 3A), the stiffness contrast images of the area show homogeneous properties with no phase separation.

Authors have to justify the absence of phase separation at 25°C, before heat treatment.

  1. Line 297

GPC procedure should be described in materials and methods.

  1. References

A significant amount of cited works are quite old and based on books. More recent papers have to be presented, mainly to support the discussion of the proposed IPN mechanism. It would be kind, if some MDPI papers are cited.

Author Response

Reviewer #2

Manuscript presents a proper structure and discussion. The following suggestion must be considered before publication:

We thank the Reviewer for this nice comment.

  1. Line 156: The blends were first heated up to 160°C to remove any internal constraints due to the solvent casting method, then cooled down to -50°C”. Experimental procedure must be described in “materials and methods”. Please remove the following sentence above “results”.

Exact. Thanks for the suggestion. This sentence has been moved in the “Materials and methods section” as proposed by the reviewer, line 114 to 115.

  1. Line 163: Between 60 and 80  phr, a second glass transition is starting to appear in the range of the resin Tg (around 80 - 120°C), indicative of a phase separation. From 80 phr, the blend Tg is relatively constant and a clear biphasic behavior was observed. This value was taken as the immiscibility threshold of the resin in SSBR. The results suggest that SBR is saturated with poly(αMSt-co-St) resin, and the excess of resin is phase-separating”. The authors claim polymer miscibility based on Tg shifting. But, they have to elucidate: why the second Tg peak is quite small, for samples >80 phr, once a large amount of performance resin was added in the formulation; the peak broadening of the first Tg as effect of resin incorporation.

The second peak reveals the contribution of the immiscible resin. For a blend composed of 100 phr of resins, it means that only 20 phr of resins contribute to this peak. In term of weight fraction, it means that this separated resin corresponds to 20/200= 10 wt% of the total weight of the tested sample and is thus rather low. Considering also the significant baseline contribution of SBR, it would explain why the intensity of the peak is so low.

  1. Line 176: When rheological measurements of SBR/poly(αMSt-co-St) were performed while heating up from 20 to 215°C, an increase of G’ was observed. This trend was not expected since SBR alone did not show such behavior (Figure 2A, red curves)”. Why crosslinking is unexpected, since in line 52, they mentioned that “commercial SBR formulations are known to be mixed with an antioxidant to prevent their thermal degradation and self-crosslinking [9]”?

Thanks for this relevant question. This is exactly what we consider as unexpected.
The antioxidant used to stabilize SBR was not removed and works well. Indeed, when heated up to 215 °C, SBR does not crosslink. As well, poly(αMSt-co-St) alone does not crosslink. However, when SBR/poly(αMSt-co-St) are blended and heated up to 215 °C, they crosslink.  

  1. Line 197: Viscoelastic properties of samples 80 and 100 phr should be presented, in Figure 2A. It would support the author´s decision to focus the experimental procedure on sample 150 phr, which was not properly described in the text.

We thank the Reviewer for this comment. The authors agree with Reviewer 2, the decision of using 150 phr loaded samples was not properly described. The decision to use 150 phr loaded sample was taken considering that the phase-separation was clearly observed as compared to 80 phr or 100 phr, to illustrate more significantly the compatibilization between SBR and poly(αMSt-co-St). A short sentence has been added in Section 3.2 to emphasize this choice, line 197 to 199.

  1. Line 212: Figure 3 shows images of SBR/poly(αMSt-co-St) taken at 25°C and sequentially  heated at 160°C at the same location. At 25°C (Figure 3A), the stiffness contrast images of the area show homogeneous properties with no phase separation. Authors have to justify the absence of phase separation at 25°C, before heat treatment.

We would like to thank the Reviewer for this relevant suggestion. That’s right it is not obvious. The absence of phase separation was attributed to the solvent casting method. Indeed, the blends were solubilized and mixed together in THF, which has a low vapor pressure. Its evaporation after disposing the samples on the plates was fast enough to avoid the phase separation of SBR and poly(αMSt-co-St), explaining the results observed by AFM. This discussion has been added to the manuscript, line 236 to 238.

  1. Line 297: GPC procedure should be described in materials and methods.

The GPC procedure is now added to the experimental section, line 127 to 141.

  1. References: A significant amount of cited works are quite old and based on books. More recent papers have to be presented, mainly to support the discussion of the proposed IPN mechanism. It would be kind, if some MDPI papers are cited.

As suggested, we added more recent publications in the list of references.

“doi.org/10.3390/polym13071116”

“doi.org/10.3390/polym12091908”

“doi.org/10.3390/polym13030352”

Reviewer 3 Report

The submitted study is interesting, well presented, valuable and has a strong practical meaning. My major comments are related to the solid state NMR analysis which could have been performed better. Also some other minor suggestions are presented below.

The structure of both SBR and poly(αMSt-co-St) should be presented in the introduction.

Line 52, usually it is not one antioxidant but a mixture of them.

As stated both in the abstract and in introduction, the work „attempts to provide an explanation” of the observed behavior. I regret that the Authors have not used any molecular modeling methods to explain this phenomenon at the molecular level. Significant improvement in this area has been done in a last few years and such theoretical methods are now accurate in predicting miscibility even for such complicated object as polymers.

Solid state NMR, why haven’t you perform CP? Further, 4s recycle delay seems to be very short, especially for Bloch decay experiments. Are you sure this was enough? By looking at the spectra, especially the one of CrR-SBR, you might have worked on saturated signal.

Figure 7, this analysis is rather ambiguous. At least some of the “peaks” of CrR-SBR are, in my opinion, only noise. Have you performed any apodization? I think that the spectrum of CrR-SBR should be registered one more time using longer recycle delay.

Conclusions could be elaborated more in a form of a separate part.

Author Response

Reviewer #3

The submitted study is interesting, well presented, valuable and has a strong practical meaning. My major comments are related to the solid-state NMR analysis which could have been performed better. Also, some other minor suggestions are presented below.

This is a very appreciated comment. Thank you very much.

  1. The structure of both SBR and poly(αMSt-co-St) should be presented in the introduction.

As suggested by the reviewer, a scheme representing the structures of SBR and poly(αMSt-co-St)   was added to the introduction part, line 67.

  1. Line 52, usually it is not one antioxidant but a mixture of them.

We would like to thank the reviewer for this insightful comment. It has been corrected accordingly.

  1. As stated both in the abstract and in introduction, the work „attempts to provide an explanation” of the observed behavior. I regret that the Authors have not used any molecular modeling methods to explain this phenomenon at the molecular level. Significant improvement in this area has been done in a last few years and such theoretical methods are now accurate in predicting miscibility even for such complicated object as polymers.

We agree with the Reviewer that molecular modeling would have been very useful to describe and predict the miscibility of the materials. As well, theoretical calculations of enthalpies of mixing could have been done. As this work is focusing on the unexpected compatibilisation of the two polymeric materials, experimental evidences are mandatory to try to give an explanation. However, we envision to publish some future works dedicated to the characterization of these blends, and the suggestion of he Reviewer will be highly considered. 

  1. Solid state NMR, why haven’t you perform CP? Further, 4s recycle delay seems to be very short, especially for Bloch decay experiments. Are you sure this was enough? By looking at the spectra, especially the one of CrR-SBR, you might have worked on saturated signal.

We actually performed the state-of-the-art CP experiments, i.e., multiCP (see more details in this pioneering paper, https://doi.org/10.1016/j.jmr.2013.11.009) for the samples. All the 13C peaks in the multiCP spectrum were also seen in the DP spectrum, so there is no saturation on the signals even with a recycle delay of 4 s in this specific case. Note that the sample was swollen by deuterated solvent in order to improve the chemical shift resolution.

Since the CP-based ssNMR techniques work only well for seeing the rigid groups of rigid samples, it could potentially miss out on the signals of the mobile groups in the swollen sample, especially some small molecules/substructures created during the “grafting” of SBR. In contrast, the DP spectrum (though the recycle delay is set short because it is not intended for quantification in this case), can capture all the 13C peaks regardless of the sample state (swollen or dry). Therefore it’s a priori more directly comparable to the solution spectrum, which is the reason we decided to show only the DP spectrum.

  1. Figure 7, this analysis is rather ambiguous. At least some of the “peaks” of CrR-SBR are, in my opinion, only noise. Have you performed any apodization? I think that the spectrum of CrR-SBR should be registered one more time using longer recycle delay.

We would like to thank the review for this comment. The spectra presented in Figure 7 were already done using apodization LB = 10 Hz. We show in the added graph (question 4) in this letter that all the peaks of CrR-SBR in the multiCP spectrum have also been observed in DP, so it is very unlikely that the 13C spins were saturated with 4 s delay. Thus, the acquisition of longer recycle delay may only improve the signal/noise ratio and according to the time required to run such experiment and that the observations may remain unchanged, we decided to respectfully decline the reviewer proposition.

  1. Conclusions could be elaborated more in a form of a separate part.

As suggested by Reviewer’s 1 and 3, we have rephrased and elaborated a separate part for the conclusion, line 67.

Round 2

Reviewer 1 Report

Minor revision

  1. Please delete scheme-1 from manuscript. It is well-known
  2. Please check self-citation, it must be below 2. 

Author Response

We would like to thank the reviewer for the comments.

However, we will respectfully decline the suggestion of deleting scheme 1. The structure of both polymers is needed to understand the molecular mechanisms behind the grafting reaction.

Concerning the self-citation, the rules of MDPI are "Authors should not engage in excessive self-citation of their own work", and not below 2. There 3 self-citations on 52 references. It represents less than 6% of the total amount of references. Furthermore, the 3 self-citation are relevant and not abusive. 

Reviewer 3 Report

The Authors have included my comments in the revised version of the manuscript.

Author Response

Thanks for the comment